# Dissociable mappings of tonic and phasic pupillary features onto cognitive processes involved in mental arithmetic

**Russell A. Cohen Hoffing**[1]*, **Nina Lauharatanahirun**[1,2], **Daniel E. Forster**[1], **Javier O. Garcia**[1,3], **Jean M. Vettel**[1,3,4], **Steven M. Thurman**[1]

**1** U.S. Combat Capabilities Development Command Army Research Laboratory, Human Research and Engineering Division, Aberdeen Proving Ground, Maryland, United States of America, **2** Annenberg School of Communication, University of Pennsylvania, Philadelphia, Pennsylvania, United States of America, **3** Department of Bioengineering, University of Pennsylvania, Philadelphia, Pennsylvania, United States of America, **4** Department of Psychological and Brain Sciences, University of California Santa Barbara, Santa Barbara, California, United States of America

* russell.a.cohenhoffing.civ@mail.mil

**Data Availability Statement:** All data files are available at an OSF public repository located at the

## Abstract

Pupil size modulations have been used for decades as a window into the mind, and several pupillary features have been implicated in a variety of cognitive processes. Thus, a general challenge facing the field of pupillometry has been understanding which pupil features should be most relevant for explaining behavior in a given task domain. In the present study, a longitudinal design was employed where participants completed 8 biweekly sessions of a classic mental arithmetic task for the purposes of teasing apart the relationships between tonic/phasic pupil features (baseline, peak amplitude, peak latency) and two task-related cognitive processes including mental processing load (indexed by math question difficulty) and decision making (indexed by response times). We used multi-level modeling to account for individual variation while identifying pupil-to-behavior relationships at the single-trial and between-session levels. We show a dissociation between phasic and tonic features with peak amplitude and latency (but not baseline) driven by ongoing task-related processing, whereas baseline was driven by state-level effects that changed over a longer time period (i.e. weeks). Finally, we report a dissociation between peak amplitude and latency whereby amplitude reflected surprise and processing load, and latency reflected decision making times.

## Introduction

Pupil features have been used for decades as a window into the mind [1]. Pupillometry captures a variety of cognitive processes including mental workload [1, 2], attention subprocesses [3], surprise [4], emotion [5], and decision making [6, 7]. Pupil size is often linked to particular behaviors or cognitive processes by examining features extracted from the time series of the pupil response [3, 8, 9]. These features can be categorized into (i) phasic, stimulus-evoked

following link https://osf.io/cs826/?view_only=
f29847cad2f84345a223032377d1d5f8.

**Funding:** The authors received no specific funding
for this work

**Competing interests:** The authors have declared
that no competing interests exist.

features such as peak amplitude and the time latency to reach peak amplitude, and (ii) tonic, non-stimulus-evoked features such as baseline pupil size. While phasic responses tend to reflect rapidly changing aspects of attention to meet immediate task demands, changes in baseline diameter tend to occur more gradually over time, reflecting generalized states of arousal (e.g., fatigue, mind wandering). A broad challenge facing the field of pupillometry from a theoretical and practical perspective, therefore, has been understanding which pupil features are most relevant for explaining variations of behavior in a particular task domain.

In the present study, we employed a classic mental arithmetic task [1, 2, 10, 11] to tease apart the relationships between pupil features and two cognitive processes involved in the task. Our study had a unique longitudinal design that included repeated measurements of mental math performance and simultaneous pupillometry over a 16-week period, including up to 8 biweekly experimental sessions per subject. For each session, we computed three pupil features (peak amplitude, peak latency, and tonic baseline) and examined within-subject effects related to mental processing load (easy versus difficult math statements) and decision-making processes (indexed by response times). Capitalizing on the longitudinal design, our analysis assessed the robustness of these relationships across multiple levels including at the single-trial level and at the within-subjects level (across multiple weeks). We used mixed effects modeling which allowed us to determine the independent influence of pupil-linked cognitive processes on mental math performance while accounting for individual differences.

We hypothesized that phasic features of the pupil response would be uniquely associated with transient cognitive processes involved in mental arithmetic processing as found in previous literature. For example, prior studies have shown greater pupil dilation during evaluation of hard compared to easy statements in a mental arithmetic task [1, 2], and other research has shown that task-evoked peak latency reflects pupil-linked decision-making processes [6–8]. Additionally, peak amplitude has been shown to be greater for incorrect compared to correct trials [9, 12, 13]. Thus, we hypothesized that peak amplitude would be modulated primarily by math question difficulty and accuracy, whereas latency would be more strongly associated with response times, irrespective of difficulty level or accuracy.

In contrast, we hypothesized that tonic features (i.e., baseline pupil size) would not be modulated by task difficulty or accuracy for individual trials, per se, but would instead reflect longer time scale variability characteristic of the response time distribution across the 8 sessions. This matches the relatively longer timescale on which baseline pupil size varies in comparison to phasic features (milliseconds/seconds). Previous research has shown that baseline pupil size is related to the rate of attentional lapses [14–16], mind wandering rates [17, 18], and performance variability [19].

Collectively, our analyses augment our theoretical understanding about the relationship between pupillometry, cognitive processes, and task performance variability. We examine the association between three pupillary features and behavioral performance at multiple time scales, and in our longitudinal sample we report robust relationships between specific pupillary features and distinct cognitive processes.

## Methods

### Participants

A total of 33 subjects were recruited to participate in the study and asked to complete 8 total sessions over the course of 16 weeks (**Fig 1A**). The data presented in this manuscript is part of a large-scale, longitudinal study protocol to understand how natural sleep history modulates the relationship between physiology and performance [20]. During the course of the study participants completed daily sleep dairies and wore actigraphy devices. We collected bi-weekly

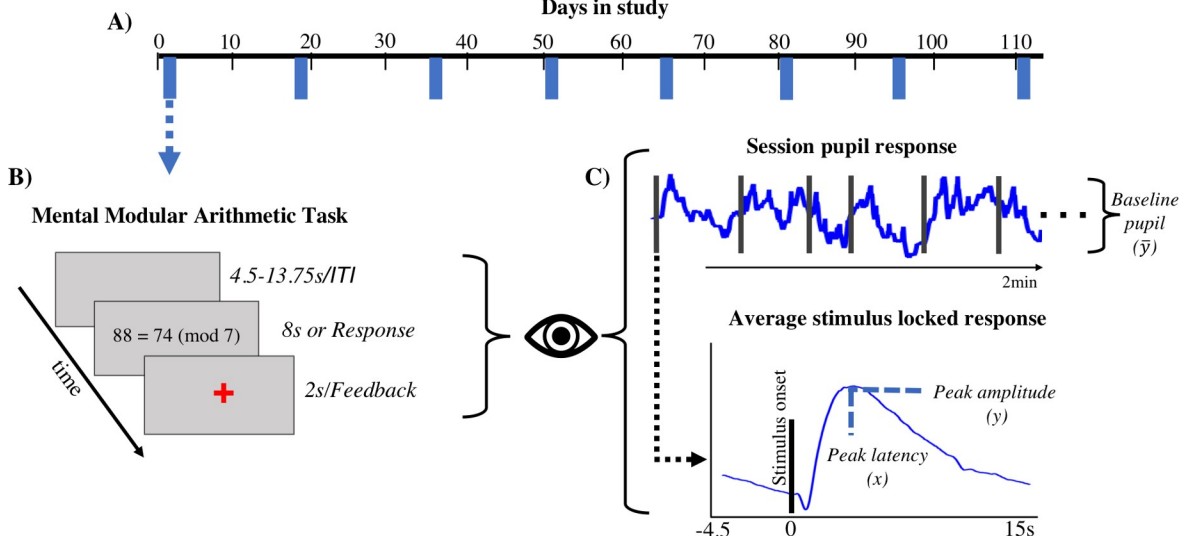

**Fig 1.** Top: **A)** Participants completed the modular mental arithmetic task on 8 separate biweekly sessions over the course of 16 weeks. **B)** During the modular mental arithmetic task participants evaluated statements by subtracting the two numbers and indicating whether the remainder was divisible by the modulo. **C)** Baseline, peak amplitude, and peak latency features were extracted from the pupil response.

neuroimaging data (i.e. EEG, MRI, fMRI), peripheral physiology (ECG), and eye tracking data while participants completed a battery of tasks in the following order: resting state, psychomotor vigilance, visual working memory, dynamic visual attention, modular arithmetic and dot probe tasks. Each session lasted approximately 120 minutes. The analyses presented in this paper specifically addressed the relationship between pupil size modulations and performance in the arithmetic task.

Our analysis was restricted to subjects that ended up completing 5 or more sessions, providing a sample size of (N = 30, 14 females, 221 total sessions). All procedures were approved by the UCSB Institutional Review Board and the accredited Institutional Review Board at US Combat Capabilities Development Command Army Research Laboratory (ARL) in compliance with the ARL Human Research Protection Program (32 Code of Federal Regulations 219 and Department of Defense Instruction 3216.01).

## Modular arithmetic task

In the modular arithmetic task participants completed 40 trials consisting of math statements [10, 11] (total trials: M = 294.66, SD = 35.60) (**Fig 1B**) in which participants were asked to subtract the two numbers and respond true if the remainder was divisible by the modulo (e.g. 7), and false if not. Difficulty was manipulated by including subtractions that involved the carrying operation (e.g. 71 $\cong$ 43 (mod 8)). Stimuli were presented for 8 seconds or until a response was made, whichever occurred first. Feedback was given to participants in the form of a green cross for correct or red cross for incorrect trials, and the cross remained on the screen for 2 seconds. The inter-stimulus-interval consisted of a blank screen randomly presented for 4.5–13.75 seconds.

## Pupillometry

Eye-tracking data were continuously recorded during the mental arithmetic task using an MR compatible EyeLink 1000 long-range eye tracker. Pupil size measurements were recorded from the left eye at a rate of 1000 Hz using the ellipse fitting mode to calculate pupil area. A

nine-point calibration procedure (built into Eyelink's software) was administered before task sessions to map eye positions to screen coordinates, in which successful calibration was determined by an average error less than 0.5 degrees [21].

Pupil data was down sampled to 250Hz. Blinks were identified and removed using cubic-spline interpolation [22]. Trial data was excluded from analysis if more than 50% of data was interpolated (M = 4.6 trials per session, SD = 4.80). Sessions were excluded from analysis if more than 20 trials were excluded (20 sessions out of 221 total sessions; mean = 0.68 sessions per person, SD = 1.02). Pupil data was epoched by time-locking to stimulus onset and normalized on a trial-by-trial basis by subtracting the pre-stimulus mean from each data point.

Analysis of pupil data was carried out by extracting three commonly identified features [8, 9] from both the session-level and trial-level pupil responses including baseline, peak amplitude, and latency of peak amplitude (Fig 1). Session-level baseline was calculated as the mean pupil size across the time series of the entire session. Trial-level baseline was calculated as the mean of un-normalized pupil data in the time window ranging from a one second prior to stimulus onset up to stimulus onset. Peak amplitude and peak latency were extracted by finding the maximum pupil dilation (and corresponding time point) in the time window ranging from stimulus onset to 10 seconds post-stimulus onset. Trials were discarded that indicated a peak latency less than 250ms post-stimulus onset or greater than 10s, amounting to removal of 8.30% of individual trials. This constraint was applied because pupil dilation responses have a minimum latency of 250ms [9], and because no response time in our study was greater than 8s. Any peak latency estimates that fell outside this window were likely due to non-stimulus or response related factors. At the session-level, peak latency and peak amplitude were extracted from the trial-averaged pupil response. Mean task-evoked pupil responses are plotted in arbitrary units (A.U.).

## Data analysis

**Behavioral analysis.** Here, we focus on response time (RT) as a measure of behavior in the mental arithmetic task. For the session-level analysis, we used median response time because the RT distribution was not normally distributed in this task. For the same reason, we computed the interquartile range (IQR) of response times from each session as a non-parametric descriptor of the breadth of the RT distribution.

**Mixed effects modeling approach.** We conducted a chi-square likelihood ratio test comparing a linear mixed model with and without random effects to determine model fit to the data. The model with random effects included an individualized intercept. We conducted a model comparison using trial-level data. These results indicated that the model with random effects provided a better fit (Trial: $\chi^2(1) = 2807$, p<0.001), demonstrating that variance estimates of the intercept (Trial: VAR = 873.38) were substantial across individuals. Consequently, all analyses employed mixed effects models to assess the relationship between pupil metrics and RT.

**Modeling behavior using linear mixed effects regression.** To investigate the relationships among pupillometry features and response time, we used mixed effects linear models to appropriately account for variation in response time across participants and across sessions. We included random effects in the model to precisely specify sources of variability at the individual level by taking into account individual variation in both initial- and session-level changes over time. Thus, this analysis allowed for robust claims of relationships between pupil predictors (i.e., baseline, peak amplitude, peak pupil) and behavioral metrics (i.e., response time, distribution of response time). We implemented the mixed effects analysis using R toolbox *lme4* and *lmer* function [23]. Regression coefficients were converted into standardized

units using the *sjstats* toolbox and *std_beta* function in R [24]. The generalized form of the statistical equation is presented below:

$$y_{it} = (\beta_0 + u_{0i}) + \beta_1 x_{1ij} + \beta_2 x_{2ij} + \beta_3 x_{3ij} + e_{ij} \tag{1}$$

In our analysis the outcome variable $y_{it}$ represents behavior (i.e. RT and RT distributions). Each predictor variable in its centered form is represented as $x_{ij}$, where 'i' represents the individual and 'j' represents session number. Here, these predictor variables represent the three pupil features (baseline, peak latency and peak amplitude) that were included in the analysis. The intercept of the regression equation is represented as $\beta_0$. The fixed effects are represented as $\beta_1, \beta_2$, and $\beta_3$. The random effects $u_{0j}$ represents the deviation from the intercept by an individual. The term $e_{ij}$ represents the residual error.

**Modeling trial-level response times.**   To assess the relationships among pupil predictors and RT on the single trial-level, we ran a mixed effects model with RT as the outcome variable. Pupil baseline, peak amplitude, and peak latency extracted from every trial were included as fixed effect variables in the model. To look at effects of condition on RT and pupil features, we ran four separate models with either RT or each of the three pupil features as outcomes. In these models, each trial was dummy coded by difficulty (easy = 0, hard = 1) and accuracy (correct = 0, incorrect = 1) and included as fixed effects in the model. In all the aforementioned models, random effects included a random intercept of subject to account for between-subjects variability. This analysis captured whether a relationship existed between the two task conditions and behavior (RT) and/or pupillary responses (baseline, peak amplitude, latency).

**Modeling session-level response time distribution.**   To assess the relationships among pupil predictors and the distribution of RT at the session-level, we ran a mixed effects model with the interquartile range (IQR) as the outcome variable. Pupil baseline, peak amplitude and peak latency were extracted from a trial-averaged pupil response across the session and included as fixed effects predictor variables in the model. Random effects included a random intercept of subject to account for between-subjects variability.

## Results

### Trial-level variation associated with peak amplitude and latency

First, we investigated whether unique associations emerge between pupillary features and behavioral variance at the single trial-level. Previous research has shown that response times vary substantially from trial-to-trial reflecting a complex interaction of fluctuations in intrinsic states (attention, engagement, boredom, etc.), task-related factors (trial difficulty, serial order effects, etc.) and individual traits (expertise, aptitude, etc.).

To investigate whether pupil features are associated with RT on the trial-level, the mixed effects model was fit with three pupil predictor variables (**Table 1**). We hypothesized that only phasic features (peak amplitude and latency) should show a significant relationship with RT on a single trial basis, and that peak latency, in particular, should better track RT variability

**Table 1.  Modeling the relationship between RT and pupil predictors at the trial-level indicate that peak latency is more strongly associated with response time suggesting that this feature maps onto decision processes that unfold over seconds.**

| Parameter | ß | SE | T | p |
|---|---|---|---|---|
| Trial Baseline | 0.081 | 0.012 | 6.933 | <0.001 |
| Trial Peak Amplitude | 0.095 | 0.012 | 9.131 | <0.001 |
| Trial Peak Latency | 0.417 | 0.010 | 43.350 | <0.001 |

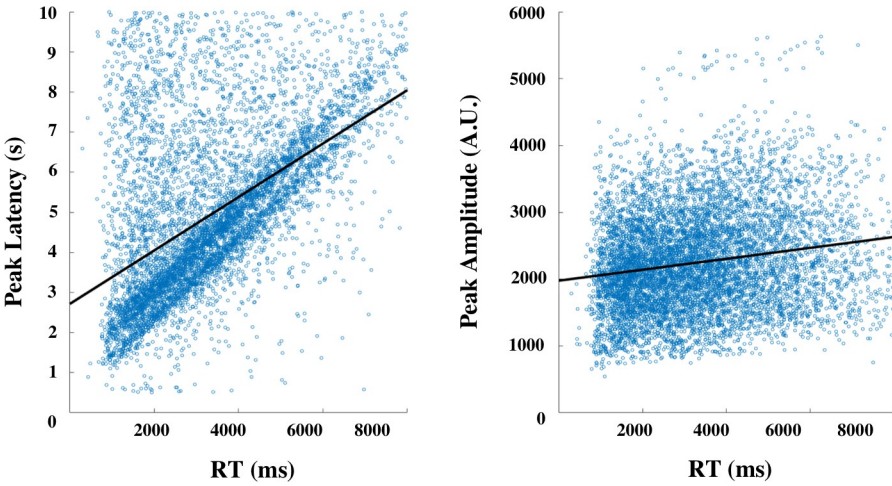

**Fig 2. Scatter plots showing response time for every individual trial in the experiment by peak latency measurements (left) and peak task-evoked amplitude measurements (in arbitrary units: A.U.) (right).** Black lines represent the least squares linear fit to the data. Peak latency, in particular, showed a striking relationship with response time, reflected in the diagonal band of data points.

due to its association with decision-related processes. In contrast with our hypothesis, all features were significantly correlated with trial-by-trial RT. However, results show that peak latency explained substantially more variance in RT in comparison to peak amplitude and baseline (T = 43.3 versus T = 9.1 and T = 6.9 respectively). Scatter plots of single trial RT and peak latency further illustrate the remarkable strength of this relationship on the single trial level (**Fig 2**).

## Peak amplitude shows an interaction with difficulty and accuracy

Previous findings suggest that difficulty (i.e., Easy, Hard) and accuracy (i.e., Correct, Incorrect; **Fig 3A**) should both influence task-evoked pupil dynamics. Difficult math questions require more intense mental processing than easier questions, and therefore should result in a larger pupil dilation response (i.e. larger peak amplitude) [1, 2, 10, 11]. Further, when feedback is provided in real-time on task performance, negative feedback for incorrect trials should also result in larger pupil dilation due to the arousal response associated with indications of failure [4]. Finally, we might expect an interaction between these two factors, whereby the pupil dilation response is increased specifically for easy questions that were answered incorrectly due to the surprise of getting an easy statement wrong [12].

Here, we evaluate the robustness of these relationships at the trial level, in contrast to a majority of prior work that has examined trial-averaged (session-level) pupil responses. To test differences in pupil features by each condition, we conducted four separate mixed effects models with RT, peak latency, peak pupil and baseline pupil as outcome variables. Difficulty and accuracy were included as fixed effects and subject was included as a random effect. First, we examined the influence of task-related factors on response time. As shown in **Fig 3B**, there was a significant interaction between difficulty and accuracy (ß = -0.097, SE = 0.014, p<0.001) as well as a main effect of both difficulty (ß = 0.224, SE = 0.010, p<0.001) and accuracy (ß = 0.169, SE = 0.013, p<0.001). The interaction effect was driven by smaller RT differences between *incorrect* easy (M = 3607ms, SD = 1351ms) and hard (M = 3828ms, SD = 1420ms) by comparison to *correct* easy (M = 2748ms, SD = 874ms) and hard (M = 3481ms, SD = 1212ms)

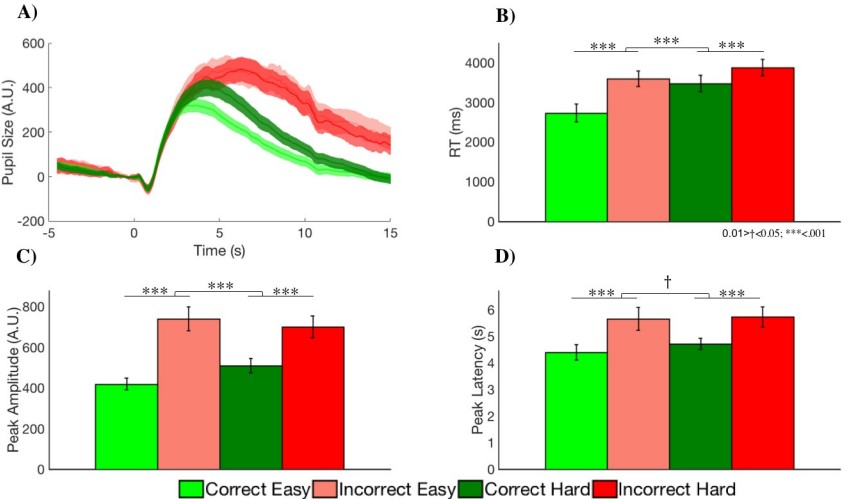

**Fig 3.** (**A**) Mean task-evoked pupil responses (in arbitrary units: A.U.) by task condition (easy, hard) and accuracy (correct, incorrect). (**B**) Bar plots illustrate how performance (mean response time) was modulated by difficulty and accuracy, demonstrating a main effect of each factor as well as an interaction effect. (**C**) Bar plots illustrate a similar pattern of results for peak amplitude of the pupil response in relation to task variables. (**D**) Peak latency was modulated by difficulty and accuracy, and showed a marginally significant interaction effect. Error bars represent standard error of the mean.

trials. In other words, while subjects were slower during hard and incorrect statements, there was an additional cost in RT associated with easy questions answered incorrectly.

Peak amplitude showed a similar pattern of results as response times (**Fig 3C**), indicating an interaction effect (ß = -0.058, SE = 0.017, p<0.001) as well as significant effects of difficulty (ß = 0.058, SE = 0.012, p<0.001) and accuracy (ß = 0.103, SE = 0.016, p<0.001). The peak amplitude interaction was driven by a smaller peak amplitude difference between incorrect easy (M = 747.39a.u., SD = 447.12a.u.) and hard (M = 670.85a.u., SD = 395.51a.u.) trials when compared to correct easy (M = 421.19a.u., SD = 244.94a.u.) and hard (M = 511.41a.u., SD = 294.14a.u.) trials. The response time cost seen in incorrect easy statements is reflected in peak amplitude, with a larger pupil dilation difference for easy incorrect and easy correct statements by comparison to the difference between hard incorrect and hard correct statements.

Peak latency also showed an interaction effect (ß = -0.040, SE = 0.017, p = 0.020) as well as significant effects of difficulty (ß = 0.113, SE = 0.012, p<0.001) and accuracy (ß = 0.190, SE = 0.016, p<0.001). The peak latency interaction was driven by a smaller peak latency difference between incorrect easy (M = 5.40s, SD = 2.24) and hard (M = 5.73s, SD = 2.20s) trials when compared to correct easy (M = 4.59s, SD = 2.06s) and hard (M = 5.04s, SD = 1.96) trials.

(**Fig 3D**). It took longer for the pupil to reach peak dilation for questions answered incorrectly compared the questions answered correctly, and there was a general trend for longer latencies for difficult versus easy questions. In comparison to the interaction found with peak amplitude, peak latency shows a marginally significant interaction that is likely driven by the strong relationship found on the trial-by-trial level.

Finally, by contrast to phasic pupil features, results of baseline pupil size indicated no significant effects of accuracy or difficulty, nor was there an interaction effect. This demonstrates that pre-stimulus pupil baseline was not associated with task difficulty, which makes sense since there was no way for the subject to anticipate the difficulty of a forthcoming trial. It also indicates that the pre-stimulus baseline state of the pupil was not diagnostic of performance on the subsequent trial.

### Baseline pupil and peak latency are associated with response time distribution

In the previous analyses, we found that baseline was unrelated to variations in trial-by-trial behavior or task difficulty. We hypothesized that tonic baseline would plausibly reflect performance over a longer time scale (e.g. across the biweekly sessions), reflecting variation in general level of activation, or arousal state, for the session as a whole. As demonstrated in the sleep deprivation literature, deficits in general arousal are likely to have a cumulative effect on the distribution of response times, in which lower arousal is often associated with increased response time variability [25], even in the absence of effects on mean response time [26, 27].

We conducted a mixed effects analysis using the IQR, a non-parametric measure of distribution breadth, as the outcome variable and pupil features (this time derived from session-averaged pupil responses) as predictor variables (**Table 2**). Results indicated that only baseline (ß = 0.166, SE = 0.067, p = 0.013) and peak latency (ß = 0.162, SE = 0.058, p = 0.005) were significantly associated with IQR, such that larger baseline pupil size and longer latency responses were indicative of a broader IQR. Thus, while baseline was not associated with RT on the single trial-level, it was related to the distribution of RT between sessions. Peak latency was found to be associated with two time scales: trial-by-trial response times and the RT distribution. This strongly suggests peak latency as a diagnostic marker of the decision making process, even showing robustness of relationship with RT variability on individual trials.

## Discussion

Pupil features have been linked to a number of cognitive processes including mental workload [1, 2], attention subprocesses [3], surprise [4], emotion [5], and decision making [6, 7]. However, there is variation across studies in terms of how the pupil-based features are evaluated as physiological correlates of cognitive processes. Thus, a practical and theoretical concern is the extent to which pupil features uniquely reflect cognitive processes since tonic and phasic aspects of the pupil response tend to reflect the integrated response of multiple brain networks over different time scales [9, 28–32].

Here, we investigated the unique relationship between a set of common pupil features (peak latency, peak amplitude, baseline) and cognitive processes (mental processing load, decision making) in a classic mental arithmetic task. To probe this question, we employed mixed effects models on data from a longitudinal pupillometry data set, allowing us to investigate whether pupil-linked cognitive processes are uniquely and robustly related to features across repeated measurements within subjects and sessions.

### Pupil latency closely tracks decision latency

In line with previous research using the mental arithmetic task and pupillometry [1, 2, 9–11], we found pupil features that were modulated by question difficulty. However, as hypothesized, not all pupil features were related to behavior in the same way. At the trial-by-trial level, peak amplitude and peak latency (but not baseline) were related to RT. In particular, both showed

**Table 2. Modeling the relationship between pupil features and IQR indicates that baseline and peak latency are associated with the distribution of response time between sessions in this longitudinal study.**

| Parameter | ß | SE | T | p |
|---|---|---|---|---|
| Session Baseline | 0.183 | 0.070 | 2.640 | 0.008 |
| Session Peak Amplitude | -0.065 | 0.060 | -1.084 | 0.278 |
| Session Peak Latency | 0.173 | 0.061 | 2.860 | 0.004 |

positive relationships with larger peak amplitude and longer peak latency being associated with slower response times. Peak latency, however, showed a striking relationship with RT at the individual trial-level (**Fig 2A**).

The robust relationship between peak latency and decision latency suggests that this pupil feature captures ongoing decisional processes. The mechanism by which this occurs is in line with the idea that the pupil begins to dilate in response to the onset of task-related processing, and continues to dilate according to mental processing demands. According to this framework, the pupil only begins to constrict once the decision process is complete, allowing the latency of the peak to robustly capture this decision making process [7, 8, 33].

In comparison to other studies, the remarkable strength of this relationship in our study may also be due to the fact that response times were long which allowed the trial-level pupil response to evolve over a 4-8s time range. However, it should be noted that one limitation of this study is that we are unable to distinguish between sub-processes driving peak latency, such as decision formation (cognitive-related) versus decision execution (motor-related). Prior work has provided evidence that perceptual decisions in a binocular rivalry task alone can drive pupillary responses [34], while other research has found that motor related processes involved in making a button press (i.e. decision, preparation, execution) can also drive the pupillary response [35]. While motor related research suggests that peak latency reflects the motor response itself, other work has found that even in the absence of an overt motor response, peak latency is still associated with the moment of decision selection [8] suggesting that the motor response may not solely be responsible for driving the correlation with peak latency in the current study. An additional potential factor is the visual presentation of feedback upon response selection, which coincided with a transient visual change following each trial that could also have induced a pupillary constriction. Since we are unable to parse out these possibilities in the current study, how these sub-processes of decision making interact with each other to influence peak latency of the pupil response remains a topic for future research.

## Peak amplitude as an indicator of processing load

While relatively much weaker than pupil latency, the relationship between peak amplitude and RT found here is in line with previous findings that peak amplitude reflects phasic attentional processes [3, 14, 36]. Complimentary research has suggested that peak amplitude may reflect an attentional pulse [37, 38] where a temporary activation of an attentional state may be needed to adapt to task demands [9, 28]. We found that modulations of peak amplitude closely paralleled effects of math question difficulty and task accuracy on behavior. Both peak amplitude and behavioral performance showed an interaction effect (**Fig 2**) whereby getting an easy question wrong induced a larger pupil dilation response, and relatively longer RT, by comparison to cases in which a difficult statement was answered incorrectly.

This result appears consistent with the interpretation of peak amplitude as a mapping of attentional state where increased attention is allocated not only during difficult trials but also during surprising events such as getting feedback that an easy question was answered incorrectly. In this case, attention is allocated to an event that engenders learning such as when a prediction error occurs. In fact, previous work has shown that incorrect trials and surprising trials are associated with increased phasic pupil responses [4, 12]. These studies have linked the noradrenaline system, involved in driving the pupil response, to the systems involved in error monitoring underlying prediction errors and learning events. In support of this interpretation, the aforementioned interaction effect between difficulty and accuracy on peak amplitude, is independent of feedback stimuli, given that the presentation of feedback stimuli was identical for difficult and easy trials. By contrast, peak latency only showed an effect of

question difficulty, where longer peak times were related to a longer RT on hard trials. This result strengthens our earlier interpretation that peak latency closely tracks the decision-making process.

## Baseline pupil diameter likely reflects longer timescale state effects on performance

We found that pre-stimulus baseline pupil size was marginally associated with RT variability on individual trials and was not associated with task-related factors like question difficulty or task accuracy. However, by capitalizing on our longitudinal design, we found that baseline pupil size was associated with variability in the breadth of the RT distribution across biweekly testing sessions. The observed correlation between pre-stimulus baseline and IQR aligns with previous research that indicates baseline pupil may reflect an overall arousal state [9, 19, 28, 39]. While baseline pupil size may not co-vary strongly with task-related factors and momentary performance within our 10-minute testing sessions, we did find that it can vary substantially from week to week in relation to global metrics of performance derived from the response time distribution. In line with this finding, research has demonstrated relationships between baseline pupil size and states like fatigue and mind wandering [18, 40].

While the relationship between baseline pupil size and performance is consistently reported in the literature, the direction of the relationship varies across studies. Our data showed a positive relationship between baseline and IQR indicating that worse performance (e.g. more variability in RTs) was associated with larger baselines, which is consistent with other studies. For example, Gilzenrat et. al. 2010 found that larger pupil diameter was related to a wider RT distribution and Franklin et al. 2013 reported it was related to increased mind-wandering [41, 42]. By contrast, studies like Unsworth et al. 2016 and van der Brink et al. 2016 reported that larger pupil diameter was related to fewer attentional lapses. This discrepancy has been noted elsewhere [15, 16, 43] and discussions have appealed to task and stimulus differences inducing different arousal states and profiles of pupillary responses. For example, tasks may cause shifts on the Yerkes-Dodson arousal curve (formalized as a quadratic), where participants show performance decrements at either end of the curve [43]. We note another possible methodological contribution to the discrepancies across studies is related to differences in methods of computing the baseline pupil feature. In our study, baseline was calculated by taking the average across the entire 10-minute session, whereas in Unsworth et al. 2016 and Gilzenrat et al. 2010 used pre-stimulus baseline measures and van der Brink et al. 2016 used a sliding window of 50 trials (approximately 40 seconds). Because of the dynamic nature of pupil-linked processes influencing the pupillary response it is possible that the baseline feature is sensitive to the temporal length of data used to extract the feature. Our analysis examined baseline pupil size on two different levels (trial-level and session-level), and our results are consistent with this notion. We found that session-level baseline was related to behavior (e.g. RT variability) but trial-level pre-stimulus baseline was not (e.g. no association with condition), suggesting that baseline over a relatively longer timescale was able to capture lower frequency state-level effects on behavior (e.g. changes that occur from session to session). This underscores the importance that studies should consider the interaction between the timescale of pupillary and behavioral change in order to better understand the underlying processes driving pupil-to-performance relationships.

## Benefits of examining pupillometry at multiple timescales

Pupillometry studies do not typically look at the relationship between the pupil response and behavior on a trial-by-trial basis; instead, pupil responses are averaged across trials to increase

the signal-to-noise ratio, an approach also used when studying event-related potentials in the EEG literature. As such, our trial-by-trial analysis results are informative about whether pupil features can actually be useful in tracking real-time cognitive processes. Despite using simple extraction methods (i.e., peak scoring) for task-evoked pupil features in this study, our results imply that (1) features may be meaningfully related to behavior on a trial-to-trial basis, and (2) given the decidedly strong association between RT and peak latency across multiple time scales in this study, from a few seconds to a few weeks, peak latency appears to be a reliable indicator of decision-making processes. This robust peak latency-to-decision latency relationship has not been reported to this extent in the literature, demonstrating the potential use in applied settings. Overall, the value in longitudinal study designs for pupillometry allows for the investigation of cognitive processes that vary over different time scales.

## Limitations and future directions

Our results focused on the unique patterns of relationships between pupil features and behavioral variance. While mixed effects modeling can provide evidence that a particular feature is sensitive to a particular behavioral variable or cognitive process, our results do not support that they are *only* driven by that process. In fact, all pupil features indicate small robust correlations with each other. One possible explanation of this correlation is that common cognitive systems co-influence the features. Previous research has found multiple neural structures that correlate with pupillary dynamics, including the locus coeruleus, superior and inferior colliculi, cholinergic basal forebrain and dopaminergic midbrain [30, 33, 44, 45]. Another possibility is that the correlation is a physical property of the pupil response itself such that features cannot be completely disentangled. For example, the rate of dilation may be near constant such that a stronger pupil dilation will tend to be associated with longer peak latency.

Nonetheless, we have shown that pupil-linked cognitive processes can be disentangled to some degree, and it helps to acquire and analyze repeated measures data from individuals using a longitudinal study design. Our study uncovered distinct relationships between behavior, task features, and pupil features at various time scales. Overall, we suggest that future pupillometry research can benefit from including many different features in their analysis to capture the role of different cognitive processes driving behavior. Research that advances the understanding of the neural underpinnings of the pupil response can help further understanding of cognitive systems because pupillometry is an easily accessible, minimally invasive, and low-cost data source. Overall, pupillometry holds promise for tracking cognitive processes and performance in naturalistic contexts, and future research must continue to advance our understanding of pupil-based features as physiological correlates of cognitive processes.

## Supporting information

**S1 Data.**
(DOCX)

## Acknowledgments

This research is aligned with the scientific aims of the Human Sciences under the United States Army and conducted as in-house research and through university affiliated research centers. We want to acknowledge the intellectual contribution of this collaborative scientific community and its strong influence on this research, with particular insight from Scott Grafton, Barry Giesbrecht, James Elliott, Piotr Franaszczuk, Scott Kerick, Brent Lance, Amar Marathe, Kaleb McDowell, Kelvin Oie, and Jon Touryan. The authors thank Gold Okafar, Alex Asturias, Phil

Beach, Mario Mendoza, Hannah Erro, and Zoe Rathbun for study coordination and subject testing.

The views and conclusions contained in this document are those of the authors and should not be interpreted as representing the official policies, either expressed or implied, of the CCDC Army Research Laboratory or the U.S. Government. The U.S. Government is authorized to reproduce and distribute reprints for Government purposes notwithstanding any copyright notation herein.

## Author Contributions

**Conceptualization:** Javier O. Garcia, Jean M. Vettel.

**Data curation:** Russell A. Cohen Hoffing.

**Formal analysis:** Russell A. Cohen Hoffing, Nina Lauharatanahirun, Daniel E. Forster.

**Methodology:** Russell A. Cohen Hoffing, Nina Lauharatanahirun, Daniel E. Forster, Javier O. Garcia, Jean M. Vettel, Steven M. Thurman.

**Project administration:** Javier O. Garcia, Jean M. Vettel.

**Supervision:** Jean M. Vettel, Steven M. Thurman.

**Visualization:** Russell A. Cohen Hoffing.

**Writing – original draft:** Russell A. Cohen Hoffing, Steven M. Thurman.

**Writing – review & editing:** Russell A. Cohen Hoffing, Nina Lauharatanahirun, Daniel E. Forster, Javier O. Garcia, Jean M. Vettel, Steven M. Thurman.

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
