## [Decision Letter · Decision Letter 0]

2 Dec 2019

PONE-D-19-28051

Dissociable mappings of tonic and phasic pupillary features onto cognitive processes involved in mental arithmetic

PLOS ONE

Dear Dr. Cohen Hoffing,

Thank you for submitting your manuscript to PLOS ONE. I have now received two reviews. Reviewer 1 prefers to remain anonymous. Reviewer 2 is Yavor Ivanov. I have also read the manuscript myself. You will be happy to learn that the reviewers are mostly positive about your work, and I agree with their assessment. They do suggest a few improvements. I also have some thoughts myself.

I invite you to respond to each of these comments in a rebuttal/ cover letter, and to revise the manuscript accordingly. Unless I feel that additional review is necessary, I will make a final decision in the next round.

Comments:

On line 339, you mention that:

> Although one possibility is that peak latency is tracking onto the motor response itself, research has found that even in the absence of an overt response, peak latency is associated with the moment of decision selection (8).

Such easy dismissal of the motor response is unsatisfying to me. Sure, peak latency *can* reflect the moment of decision, but that does not mean that it actually does in your case. To me, the strength of the peak latency v RT correlation suggests that a motor response is part of what drives this correlation. And it's not just the motor response either: the fact that the display changes at the moment of response means that there is a visual change, causing pupil constriction, which is locked to the response. This is not a problem per se (and nothing can be done about it now, in any case), but it's something that warrants additional discussion.

As another point, you're taking session into account as a random slope. This implies that you're not interested in how behavior and pupil responses change systematically over time, because those effects will be captured by the by-session random slope. But why not? What is the theoretical motivation behind your statistical models? This point is related to that of Reviewer 1, who would like to see more methodological details.

We would appreciate receiving your revised manuscript by Jan 16 2020 11:59PM. To enhance the reproducibility of your results, we recommend that if applicable you deposit your laboratory protocols in protocols.io, where a protocol can be assigned its own identifier (DOI) such that it can be cited independently in the future. For instructions see: http://journals.plos.org/plosone/s/submission-guidelines#loc-laboratory-protocols

We look forward to receiving your revised manuscript.

Kind regards,

Sebastiaan Mathôt, Ph.D.

Academic Editor

PLOS ONE

Journal Requirements:

'This research is aligned with the scientific aims of the Human Sciences research funded by the United States Army as both in-house research and through university affiliated research centers.

'The authors received no specific funding for this work'

Additional Editor Comments (if provided):

I have now received two reviews. Reviewer 1 prefers to remain anonymous. Reviewer 2 is Yavor Ivanov. I have also read the manuscript myself. You will be happy to learn that the reviewers are mostly positive about your work, and I agree with their assessment. They do suggest a few improvements. I also have some thoughts myself.

I invite you to respond to each of these comments in a rebuttal/ cover letter, and to revise the manuscript accordingly. Unless I feel that additional review is necessary, I will make a final decision in the next round.

Comments:

On line 339, you mention that:

> Although one possibility is that peak latency is tracking onto the motor response itself, research has found that even in the absence of an overt response, peak latency is associated with the moment of decision selection (8).

Such easy dismissal of the motor response is unsatisfying to me. Sure, peak latency *can* reflect the moment of decision, but that does not mean that it actually does in your case. To me, the strength of the peak latency v RT correlation suggests that a motor response is part of what drives this correlation. And it's not just the motor response either: the fact that the display changes at the moment of response means that there is a visual change, causing pupil constriction, which is locked to the response. This is not a problem per se (and nothing can be done about it now, in any case), but it's something that warrants additional discussion.

As another point, you're taking session into account as a random slope. This implies that you're not interested in how behavior and pupil responses change systematically over time, because those effects will be captured by the by-session random slope. But why not? What is the theoretical motivation behind your statistical models? This point is related to that of Reviewer 1, who would like to see more methodological details.

Reviewers' comments:

Reviewer's Responses to Questions

**Comments to the Author**

1. Is the manuscript technically sound, and do the data support the conclusions?

Reviewer #1: Yes

Reviewer #2: Yes

2. Has the statistical analysis been performed appropriately and rigorously? 

Reviewer #1: Yes

Reviewer #2: Yes

3. Have the authors made all data underlying the findings in their manuscript fully available?

Reviewer #1: Yes

Reviewer #2: Yes

4. Is the manuscript presented in an intelligible fashion and written in standard English?

Reviewer #1: Yes

Reviewer #2: Yes

5. Review Comments to the Author

Reviewer #1: Review of “Dissociable mappings of tonic and phasic pupillary features onto cognitive processes involved in mental arithmetic”

Summary: The authors present a study examining both tonic and phasic pupillary correlates during a mental arithmetic task. Participants performed a mental arithmetic task varying in difficulty over serval days and weeks while pupillary responses were recorded. The results suggested that trial-to-trial variation in RT was related to peak latency and to a lesser extent peak amplitude. Examining difficulty and accuracy suggested an interaction such that incorrect answers had larger amplitudes than correct responses, and when a correct response was given there was an effect of difficulty (larger amplitude for harder items). Finally, baseline pupil was associated with aspects of the RT distribution. The authors suggest that it is important to examine both tonic and phasic aspects of pupillary responses during cognitive tasks.

This is an interesting paper that examines the importance of both tonic and phasic pupillary responses during mental arithmetic. The writing and results are generally clear and consistent with the authors’ interpretations. I think the paper should be published following some minor revisions.

First, I think a bit more information is needed in the methods, or aspects of the methods just need to be a bit clearer. For example, it looks like participants performed a battery of other tasks in each session and perhaps this was associated with a sleep study. Is that correct? A bit more info on when the math task occurred in the battery and the overall reason for collecting the current data is needed. What is the rationale for N = 33. Is this based on an a priori power analysis? Additionally, it looks like participants completed 40 trials per session (8 total sessions) for 320 total trials per participant. Is that correct? Please be sure to really spell out the full methods in detail so that the reader knows what is going on rather than trying to piece it together throughout the methods.

Second, the finding that error responses have larger amplitudes than correct responses is consistent with prior research demonstrating and error pupillary response as noted in the paper. However, the authors seem to suggest that this is due to the feedback provided. But, it looks like the peak response happens around 5 s which is sooner than when the feedback is provided. Thus, this pupillary response seems consistent with overall error monitoring processes independent of feedback.

Finally, please provide the a bit more information on the mixed effects models. Perhaps include the overall model equations so that the reader knows exactly what effects are in the models.

Reviewer #2: This study is a longitudinal investigation of within- and between-subject variability in three pupil size measures: peak dilation, peak latency, and baseline size. In this study, the researchers use a mental arithmetic task with two difficulty conditions and measure trial-to-trial and session-to-session variation in pupil size. The researchers argue that response times (RTs) in this task index mental effort and decision-making. On a trial-to-trial level, they report that both peak dilation and peak latency have a significant positive relationship with RTs, whereas baseline pupil size does not. On a session-to-session level, they report that RT variability is correlated with peak dilation and baseline size.

The study is methodologically sound, and it furthers our knowledge of widely used psycho-physiological measures. Crucially, no studies have investigated the relationship between pupil size measures and decision-making on such a timescale.

The introduction is clearly written and to the point, and the hypotheses and expected results are clearly stated. The methods are well described, and the pre-processing and analyses are conducted and reported with sufficient rigor.

I have one comment regarding the discussion. The present results suggest that pupil dilation and RT variability are positively correlated. In section “Baseline Pupil and Peak Latency are Associated with RT Distribution”, the authors mention that research on sleep deprivation suggests that lower arousal is linked with greater RT variability. However, in the literature on pupillometry higher levels of arousal are often actually linked to larger pupil dilation, going against the presented results. Therefore, the results of this study go against the current understanding of the relationship between arousal and pupil size, which I think is an interesting finding and could be emphasized more.

6. PLOS authors have the option to publish the peer review history of their article (what does this mean?). If published, this will include your full peer review and any attached files.

Reviewer #1: No

Reviewer #2: Yes: Yavor Ivanov

---

## [Author Response · Author response to Decision Letter 0]

14 Jan 2020

Please see enclosed document titled Response to Reviewers where all comments have been addressed.

---

## [Decision Letter · Decision Letter 1]

4 Feb 2020

PONE-D-19-28051R1

Dissociable mappings of tonic and phasic pupillary features onto cognitive processes involved in mental arithmetic

PLOS ONE

Dear Dr. Cohen Hoffing,

Thank you again for your submission to PLoS ONE. As you will  read, the reviewers are mostly satisfied with how you addressed their comments. However, as I was trying to understand the comment of R2 about the covariates, I became a confused about the analysis that is described on L255 - 266. You describe this as a repeated measures ANOVA with subject as a random effect on trial-level data. To me, this sounds contradictory: either you conduct a RM-ANOVA on aggregated data, or you conduct a LMER on trial level data. The large degrees of freedom also suggest that maybe something went wrong here. (Specifically, I'm thinking that you may have entered trial-level data into a RM-ANOVA test that does not take this into account.)

Could you therefore briefly (but clearly) clarify what kind of statistical test was conducted here, and convince me that this is statistically valid? (I'm not a statistician myself, but I'll do my best and possibly consult someone who is.)

(Standard email below)

Thank you for submitting your manuscript to PLOS ONE. After careful consideration, we feel that it has merit but does not fully meet PLOS ONE’s publication criteria as it currently stands. Therefore, we invite you to submit a revised version of the manuscript that addresses the points raised during the review process.

We would appreciate receiving your revised manuscript by Mar 20 2020 11:59PM. To enhance the reproducibility of your results, we recommend that if applicable you deposit your laboratory protocols in protocols.io, where a protocol can be assigned its own identifier (DOI) such that it can be cited independently in the future. For instructions see: http://journals.plos.org/plosone/s/submission-guidelines#loc-laboratory-protocols

We look forward to receiving your revised manuscript.

Kind regards,

Sebastiaan Mathôt, Ph.D.

Academic Editor

PLOS ONE

Reviewers' comments:

Reviewer's Responses to Questions

**Comments to the Author**

1. If the authors have adequately addressed your comments raised in a previous round of review and you feel that this manuscript is now acceptable for publication, you may indicate that here to bypass the “Comments to the Author” section, enter your conflict of interest statement in the “Confidential to Editor” section, and submit your "Accept" recommendation.

Reviewer #1: All comments have been addressed

Reviewer #2: All comments have been addressed

2. Is the manuscript technically sound, and do the data support the conclusions?

Reviewer #1: Yes

Reviewer #2: Yes

3. Has the statistical analysis been performed appropriately and rigorously? 

Reviewer #1: Yes

Reviewer #2: Yes

4. Have the authors made all data underlying the findings in their manuscript fully available?

Reviewer #1: Yes

Reviewer #2: Yes

5. Is the manuscript presented in an intelligible fashion and written in standard English?

Reviewer #1: Yes

Reviewer #2: Yes

6. Review Comments to the Author

Reviewer #1: Review of “Dissociable mappings of tonic and phasic pupillary features onto cognitive processes involved in mental arithmetic”

Summary: The authors present a study examining both tonic and phasic pupillary correlates during a mental arithmetic task. Participants performed a mental arithmetic task varying in difficulty over serval days and weeks while pupillary responses were recorded. The results suggested that trial-to-trial variation in RT was related to peak latency and to a lesser extent peak amplitude. Examining difficulty and accuracy suggested an interaction such that incorrect answers had larger amplitudes than correct responses, and when a correct response was given there was an effect of difficulty (larger amplitude for harder items). Finally, baseline pupil was associated with aspects of the RT distribution. The authors suggest that it is important to examine both tonic and phasic aspects of pupillary responses during cognitive tasks.

The authors have addressed all of my concerns and I recommend acceptance.

Reviewer #2: I think the authors have sufficiently addressed our previous comments.

I have two minor remarks:

- on line 171 you repeated "a better" twice in a row

- As far as I understand, on lines 202-204 you mention that difficulty and accuracy were included as covariates in the previously described mixed effects model (with RTs as DV and pupil features as IVs). Later in the results, paragraph at line 255, you report a RM-ANOVA with difficulty and accuracy as IVs, and their effects as covariates are not reported. The RM-ANOVA results are of course valid. But it just felt a bit inconsistent to first introduce these factors as part of the mixed model, but then report the results from a separate ANOVA.

7. PLOS authors have the option to publish the peer review history of their article (what does this mean?). If published, this will include your full peer review and any attached files.

Reviewer #1: No

Reviewer #2: No

---

## [Author Response · Author response to Decision Letter 1]

26 Feb 2020

The response to reviewers is included in the attached file "Response to Reviewers".

---

## [Editor Report · Decision Letter 2]

3 Mar 2020

Dissociable mappings of tonic and phasic pupillary features onto cognitive processes involved in mental arithmetic

PONE-D-19-28051R2

Dear Dr. Cohen Hoffing,

It is my pleasure to accept your manuscript for publication is PLoS ONE. Thank you for your contribution, and also for working with the reviewers and me on this manuscript. Congratulations!

This means that your manuscript has been judged scientifically suitable for publication and will be formally accepted for publication once it complies with all outstanding technical requirements. Within one week, you will receive an e-mail containing information on the amendments required prior to publication. When all required modifications have been addressed, you will receive a formal acceptance letter and your manuscript will proceed to our production department and be scheduled for publication.

With kind regards,

Sebastiaan Mathôt, Ph.D.

Academic Editor

PLOS ONE
---

## [Editor Report · Acceptance letter]

9 Mar 2020

PONE-D-19-28051R2 

Dissociable mappings of tonic and phasic pupillary features onto cognitive processes involved in mental arithmetic 

Dear Dr. Cohen Hoffing:

I am pleased to inform you that your manuscript has been deemed suitable for publication in PLOS ONE. Congratulations! Your manuscript is now with our production department. 

With kind regards,

on behalf of

Dr. Sebastiaan Mathôt 

Academic Editor

PLOS ONE